# Improving Resistance of Mango to *Colletotrichum gloeosporioides* by Activating Reactive Oxygen Species and Phenylpropane Metabolism of *Bacillus amyloliquefaciens* GSBa-1

**DOI:** 10.3390/metabo14080417

**Published:** 2024-07-29

**Authors:** Wenya Li, Hua Chen, Jianhu Cheng, Min Zhang, Yan Xu, Lihua Wang, Xueqiao Zhao, Jinyao Zhang, Bangdi Liu, Jing Sun

**Affiliations:** 1School of Architecture and Art, Hebei University of Engineering, Handan 056038, China; 2Academy of Agricultural Planning and Engineering, Ministry of Agriculture and Rural Affairs, Beijing 100125, China; 3Key Laboratory of Agro-Products Primary Processing, Ministry of Agriculture and Rural Affairs of China, Beijing 100125, China; 4School of Food and Pharmacy, Zhejiang Ocean University, Zhoushan 316022, China

**Keywords:** *Mangifera indica*, antagonistic bacteria preservation, resistance to anthracnose, reactive oxygen species, phenylpropanoid metabolic pathway

## Abstract

This study aimed to explore the effects of *Bacillus amyloliquefaciens* GSBa-1 treatment on anthracnose disease resistance and the metabolism of reactive oxygen species (ROS) and phenylpropanoids in mangoes during storage. Mangoes were soaked in a solution containing 1 × 10^8^ CFU/mL of *B. amyloliquefaciens* GSBa-1. The anthracnose disease incidence, disease index, respiration intensity, ethylene release, reactive oxygen species content, and the activities of related metabolic enzymes, phenylpropanoid-related metabolic enzymes, and phenolic acids in the skin and pulp of mangoes were investigated under normal temperature storage conditions. The results showed that the antagonistic bacterial treatment (ABT) did not significantly inhibit the growth of *Colletotrichum gloeosporioides* in vitro. However, it significantly reduced the incidence of mango anthracnose disease when applied to the mango peel. ABT enhanced the latent resistance of mango to anthracnose disease by activating its reactive oxygen and phenylpropanoid metabolism. It maintained higher levels of ROS production and elimination in the peel. Moreover, it rapidly activated manganese superoxide dismutase, induced the accumulation of H_2_O_2_, and enhanced the activity of manganese superoxide dismutase, catalase, ascorbate peroxidase, and peroxidase in the mango peel. Furthermore, ABT activated phenylalanine ammonia-lyase, cinnamic acid-4-hydroxylase, 4-coumaroyl-CoA ligase, and cinnamyl alcohol dehydrogenase in the mango peel and pulp, promoting the accumulation of antifungal phenolic acids such as gallic acid, catechins, and ellagic acid. *Bacillus amyloliquefaciens* GSBa-1 may be a potent inhibitor of mango anthracnose, primarily enhancing the resistance of mangoes to anthracnose by synergistically activating ROS in the peel and phenylpropanoid metabolism in the pulp, thereby reducing the incidence of anthracnose effectively.

## 1. Introduction

*Mangifera indica* L., commonly known as mango, is the second-most produced tropical fruit globally after ananas [1]. It is expected to reach a production volume of 54.83 million tons in 2020. Nevertheless, the post-harvest losses of mangoes remain relatively high, with a global average loss rate exceeding 30%. In some regions with inadequate cold-chain preservation, loss rates reach 60% or higher. Among the factors contributing to these losses, pathogenic infections leading to fruit decay are the most significant [2]. *Colletotrichum* spp. are the most common pathogens responsible for the deterioration of mangoes. These pathogens cause anthracnose, a fungal disease characterized by prominent, sunken, deep brown or black lesions in mature mangoes, which trigger a series of senescence and decay [3]. *Colletotrichum gloeosporioides* typically invades the flower buds of mangoes during the flowering season, remaining dormant until the mangoes mature. When mangoes reach maturity, the fungus becomes active, promoting rapid aging and rotting of the fruit. Effective prevention and management of anthracnose during storage is currently one of the most challenging issues in mango horticulture.

Current methods for suppressing mango anthracnose primarily involve fungicidal agents such as triazoles (e.g., tebuconazole), benzimidazoles (e.g., benomyl), and strobilurins (e.g., azoxystrobin). However, increasing global fungal resistance to these traditional fungicides has led to a gradual decline in their effectiveness. Moreover, the residues of these chemicals have adverse effects on the environment and human health [4]. In addition, some consumers prefer fruit products preserved without chemical fungicides. Because of multiple factors such as market demand and technological advancements, alternative physical and biological methods for controlling mango anthracnose have emerged, including chitosan coatings for preservation [5], low-temperature storage [6], and the application of natural extracts [7]. Biocontrol is a novel approach which primarily involves the use of antagonistic microorganisms, including bacteria, fungi, and actinomycetes, to combat pathogenic microorganisms in fruits and vegetables. Wang et al. [8] isolated *Bacillus velezensis* Yb-1 from healthy cucumber leaves for the antagonistic control of anthracnose in chili peppers and tomatoes. This strain significantly inhibited the mycelial growth of *Colletotrichum capsici* in chilies and gray mold in tomatoes. Additionally, *Bacillus amyloliquefaciens* GSBa-1 (CGMCC No.13745), previously isolated from traditional Chinese liquor fermentation starters, possesses high antioxidant activity [9]. In our previous study, we found that *B. amyloliquefaciens* GSBa-1 effectively enhanced the benzenepropanoid metabolism of citron fruit, thereby slowing its aging process [10]. It also effectively inhibited post-harvest diseases in mangoes and dates, thereby reducing the incidence of fruit disease [11]. However, its specific mechanism of action in disease prevention requires further study to facilitate its application as a biocontrol agent across a wide variety of fruits and vegetables.

The primary mechanisms proposed for the action of biocontrol agents in fruit preservation can be categorized into five classes: nutrient and spatial competition, secretion of antifungal antibiotics, release of volatile metabolites, enhancement of host resistance, and hyperparasitism [12]. Induced resistance is a major mode of action of many biocontrol agents for fruits and vegetables. Zhou et al. [13] found that *Debaryomyces nepalensis*, a yeast strain from Nepal, can induce the activity of various defense enzymes in mangoes, thereby enhancing their defense capacities and controlling the incidence of anthracnose. Therefore, we hypothesized that *B. amyloliquefaciens* GSBa-1 might also enhance disease resistance in mangoes, thereby reducing the occurrence of anthracnose. The phenylpropane pathway is a key metabolic pathway that plays a critical role in fruit disease resistance by promoting the accumulation of resistant compounds such as phenolic acids, flavonoids, and anthocyanins, which can enhance fruit resistance against pathogenic microorganisms [10]. The metabolism of reactive oxygen species (ROS) is another major metabolic pathway in fruits and vegetables that combats the invasion of exogenous microorganisms. ROS also serve as key post-harvest signaling molecules, which rapidly activate the phenylpropane pathway and other defense mechanisms. This induction enhances fruit resistance and mitigates storage-related losses [14]. You et al. [15] showed that *B. siamensis* could significantly inhibit disease indexs and browning index on mango and litchi fruits, which was improved the resistance of fruit to disease.

To investigate the potential of *B. amyloliquefaciens* GSBa-1 as an antagonistic bacterium for the inhibition of fruit diseases, we selected mango as the experimental sample as it is susceptible to anthracnose infection during storage. We applied *B. amyloliquefacien* GSBA-1 as a coating on the mango surface to observe decay and anthracnose incidence during storage. Additionally, we cultured *B. amyloliquefacien* GSBA-1 with *C. gloeosporioides* in vitro to observe the inhibitory effect of *B. amyloliquefacien* GSBA-1 on this anthracnose-causing fungus. We also explored the influence of *B. amyloliquefaciens* GSBa-1 on mango ROS and phenylpropanoid metabolic pathways. We aimed to elucidate the mechanisms by which *B. amyloliquefaciens* GSBa-1 maintains mango quality and reduces anthracnose incidence, providing a foundation for its application as a biocontrol agent in the post-harvest storage and preservation of mangoes and other fruits and vegetables.

## 2. Materials and Methods

### 2.1. Materials

*M. indica* Tainong No. l mangoes, harvested in 2023 in Baise, Guangxi, China, were used in this study. The fruits were harvested at the production site and transported to the laboratory within 24 h. The entire transportation process was carried out using a cold-chain system. Upon arrival at the laboratory, fruit samples were equilibrated for 12 h before experiments were performed. *B. amyloliquefaciens* GSBa-1, a biocontrol agent, was isolated from traditional liquor starters at the Dairy Laboratory of Beijing Technology and Business University [16]. Before use, the strain was stored under freezing conditions and activated for 12 h before the immersion treatment. *C. gloeosporioides* (strain number: ACCC 36814) was isolated from a naturally diseased *M. indica* Tainong No. l mango. It was purchased from the Institute of Microbiology, Chinese Academy of Sciences, for in vitro antibacterial activity tests in this study.

All reagents used in the experiment, including sodium hydroxide, phenolphthalein indicator, and concentrated sulfuric acid, were of analytical grade and purchased from Shanghai Maclyn Biochemical Technology Co., Ltd., Shanghai, China. The hydrogen peroxide assay kit and the test kit for superoxide anion free radical production and inhibition were obtained from Nanjing Jiacheng Bioengineering Research Institute Co., Ltd., Nanjing, China. Potato dextrose agar (PDA) liquid medium was procured from China National Pharmaceutical Group Chemical Reagent Co., Ltd., Shanghai, China.

### 2.2. Antagonistic Treatment Method

The *B. amyloliquefaciens* GSBa-1 culture suspension was prepared as described by Wang et al. [17]. Activated *B. amyloliquefaciens* GSBa-1 was inoculated into conical flasks containing PDA liquid medium. The flasks were incubated at 30 °C while shaking at 150 rpm for 24 h, resulting in a culture suspension of 1 × 10^8^ colony-forming units per mL (CFU/mL).

Uniform-sized, mechanically undamaged, and disease-free mangoes were selected for the experiment. Two treatment groups were established: an antagonistic bacteria treatment (ABT) group and a control (CK) group. The specific treatment methods were as follows:

ABT group: Forty fresh mangoes were soaked in 36 L of *B. amyloliquefaciens* GSBa-1 suspension (containing potato dextrose agar culture medium) for 15 s for coating. The mangoes were removed and drained until no droplets remained on the fruit skin. The processed fruits were laid flat on a sterile plastic tray, with sterile paper towels placed underneath the tray. The samples were placed flat in individual fruit and vegetable storage rooms for further room temperature storage experiments. During room temperature storage, the chamber was maintained at a humidity level of 80 ± 0.5% and a storage temperature of 25 ± 1 °C.

CK group: Forty fresh mangoes were immersed in PDA culture medium for 15 s, removed, and drained until no droplets remained on the peel surface. The storage conditions were identical to those used in the ABT group.

Throughout the experiment, mango samples were collected on days 0, 4, 8, and 12. Each sample consisted of 10 mangoes, and quality measurements were conducted. The measurements were repeated three times for both groups, and the results were averaged.

### 2.3. Determination of Physical and Chemical Indices of Storage

#### 2.3.1. Incidence and Disease Index Statistics

Referring to the method of Jiang et al. [18], the mangoes in each group were classified into five grades based on disease spot area: grade 0 (no diseased spot), grade 1 (diseased spot area < $\frac{1}{10}$), grade 2 (diseased spot area accounted for $\frac{1}{10}$~$\frac{1}{5}$), grade 3 (diseased spot area accounted for $\frac{1}{5}$~$\frac{1}{2}$), and grade 4 (diseased spot area > $\frac{1}{2}$). The disease index (DI) was calculated using the following formula:$$DI=\frac{\sum \left(number\;of\;fruits\;in\;each\;disease\;grade\;\times \;the\;disease\;grade\right)}{\left(total\;number\;of\;investigations\;\times \;the\;highest\;disease\;grade\right)}$$

#### 2.3.2. In Vitro Anthracnose Inhibition Tests

We conducted in vitro analyses to verify whether *B. amyloliquefaciens* GSBa-1 can effectively inhibit the growth of *C. gloeosporioides* as an antagonistic bacterium. Briefly, *C. gloeosporioides* mycelium was collected using a sterile inoculation loop and inoculated onto the central part of PDA plates, followed by incubation at 30 °C for 48 h. Two rings of the mycelium were scraped from the well-grown plate and suspended in physiological saline to achieve a concentration of 1 × 10^8^ CFU/mL. The *B. amyloliquefaciens* GSBa-1 suspension was prepared as described in Section 2.2.

For the CK group, 150 µL of *C. gloeosporioides* suspension was inoculated onto PDA plates, followed by incubation at 32 °C for 48 h. For the ABT group, 150 µL of *C. gloeosporioides* suspension and 2 µL of *B. amyloliquefaciens* GSBa-1 suspension were inoculated onto PDA plates. The plates were incubated at 32 °C for 48 h. After incubation, the microbial growth was observed, and images were taken.

#### 2.3.3. Respiratory Strength and Ethylene Release

Respiratory intensity and ethylene release were assessed according to the methods of Zhou et al. [19], using an FX950 ethylene analyzer (Beijing Sunshine billion Star Technology Co., LTD., Beijing, China). Ethylene release content and respiratory intensity were calculated and expressed as mg·kg^−1^·h^−1^.

#### 2.3.4. Determination of Hydrogen Peroxide and Superoxide Anion Free Radicals

The contents of hydrogen peroxide (H_2_O_2_) and superoxide anion radical ($O_{2}^{\cdot -}$) were determined according to the methods of Xin et al. [20] and Xin et al. [14]. To determine H_2_O_2_ content, 2 g of frozen mango powder was weighed in a 10 mL centrifuge tube. Subsequently, 5 mL of 0.1 mol/L phosphate buffer solution (pH 7.4) was added and mixed. The suspension was centrifuged (5180× *g*) at 4 °C for 20 min, and the supernatant was collected. The corresponding reagent was added to the supernatant according to the kit instructions. The absorbance was measured at a wavelength of 405 nm. H_2_O_2_ content was expressed as mmol/g.

To determine $O_{2}^{\cdot -}$ content, 2 g of frozen mango powder was weighed in a 10 mL centrifuge tube. Subsequently, 5 mL of 0.05 mol/L phosphate buffer (pH 7.8) containing 0.001 mol/L of ethylenediamine tetraacetic acid, 0.3% TritonX-100, and 2% polyvinylpyrrolidone was added and mixed. After centrifuging at 10,000 r/min and 4 °C for 20 min, the supernatant was collected. The corresponding reagents were added to the supernatant according to the instructions of the determination kit. Subsequently, the absorbance was measured at a wavelength of 550 nm. The $O_{2}^{\cdot -}$ content was expressed as U/g.

#### 2.3.5. Determination of Enzyme Activities Related to ROS Metabolism

The 2 g sample was placed in a 10 mL centrifuge tube, mixed with 5 mL of 0.1 mol/L of phosphate buffer (pH 7.5) containing 0.005 mol/L of dithiothreitol and 5% polyvinylpyrrolidone. After centrifugation (14,000× *g*) at 4 °C for 30 min, the supernatant was collected. Peroxidase (POD), catalase (CAT), superoxide dismutase (SOD), and ascorbate peroxidase (APX) activities were detected using kits from Nanjing Jiancheng Bioengineering Institute and measured by spectrophotometry and colorimetry (UV–2100UNICO, Shanghai Chemical Experimental Equipment, Shanghai, China). Enzymatic activity was expressed as U/g.

#### 2.3.6. Determination of Enzyme Activities Related to Phenylpropane Metabolism

The activities of phenylalanine ammonia lyase (PAL), cinnamic acid-4-hydroxylase (C4H), 4-coumarate-CoA Ligase (4CL), and p-coumarate 3-hydroxylase (C3H) were determined by spectrophotometry according to the method of Jiang et al. [21].The enzymes were extracted by homogenizing 1.0 g of sample powder (ground in liquid nitrogen) in 3 mL of TRIS-HCl buffer (pH = 8.8, containing 40 g/L PVP), 15 mmol/L of β-mercaptoethanol, 10% methylene, and 2% (*w*/*v*) polyethylene glycol in an ice bath. After 30 min, the mixture was centrifuged at 12,000× *g* for 30 min (4 °C), and the supernatant (crude enzyme solution) was collected. The reaction mixture consisted of 0.2 mL of crude enzyme solution and 0.8 mL of reaction solution (10 mmol/L of nicotinamide adenine dinucleotide phosphorus (NADP) and 5 mmol/L of trans-cinnamic acid). The mixture was incubated in a water bath at 37 °C for 30 min. The reaction was stopped with 1 mol/L of HCl and centrifuged if precipitation occurred. The absorbance was measured at 400 nm and compared to that of 0.2 mL of PBS and 0.8 mL of reaction solution. Enzyme activity was expressed as U/g FW.

#### 2.3.7. Determination of Phenolic Compounds by HPLC

Phenolic compounds in mango peel and pulp were determined using high-performance liquid chromatography coupled with mass spectrometry, based on the method of Villamil et al. [22] with slight modifications. To prepare the extract, 5 g of mango skin and pulp were ground in liquid nitrogen, placed in a centrifuge tube (50 mL), and mixed with 20 mL of 70% ethanol solution. The mixture was subjected to ultrasonic treatment for 1 h, followed by centrifugation (14,000× *g*) at 4 °C for 20 min. The supernatant was collected as the extract. HPLC was performed under the following conditions: 0–15 min, 12–25% B; 15–25 min, 25–35% B; 25–50 min 35–55% B; 50–60 min, 55–65% B; 60–70 min 65–12% B. The absorbance was measured at a wavelength of 280 nm. A laboratory-prepared mixed standard was used to qualitatively and quantitatively analyze the phenols in the mango tissue. The content of phenolic compounds was expressed as mg/kg.

### 2.4. Statistical Analysis

Data were processed and analyzed using Microsoft Excel 2019, including calculation of the mean, standard deviation (SD), and graph generation. All experiments were conducted in triplicate, and results were expressed as mean ± SD. Significant differences between groups were analyzed using two-factor analysis of variance (ANOVA) in SPSS 26.0 software. Pearson’s correlation coefficient was used to analyze correlations between ROS and phenylpropanoid metabolism enzymes and compounds, and results were presented in a circular heatmap matrix. Statistical significance was set at *p* < 0.05.

## 3. Results

### 3.1. Impact of ABT on the Incidence of Mango Anthracnose during Storage and In Vitro

Figure 1 shows the effects of ABT on mango respiration rate, ethylene production, disease incidence, disease index, and its inhibitory effect on mango anthracnose in vitro. Throughout the storage period, ABT significantly reduced all these parameters (*p* < 0.05). Respiration intensity and ethylene production remained consistently lower in the ABT group compared to the CK group. Consequently, ABT effectively slowed the mango ripening process and maintained better post-harvest quality.

The decay incidence and index remained lower for mangoes in the ABT group than those in the CK group throughout the 12-day storage period. By the 12th day of storage, the decay incidence in the CK group reached 86.67%, while in the ABT group, it was only 66.67%. Figure 1E shows that by the 8th day, mangoes in the CK group had visible anthracnose spots on the peel, which expanded significantly by the 12th day, with clear anthracnose symptoms in the pulp. Conversely, the mangoes in the ABT group had fewer and smaller anthracnose spots by the 12th day of storage. These results suggest that ABT provides continuous antifungal effects during prolonged storage. This may be attributed to the antagonistic action of *B. amyloliquefaciens* GSBa-1 against *C. gloeosporioides* [23]. Thus, ABT can suppress mango anthracnose, thereby delaying fruit ripening and senescence.

To verify whether the inhibition of *C. gloeosporioides* growth by ABT was due to direct inoculation of *B. amyloliquefaciens* GSBa-1 on the mango peel through spatial nutrient competition, an in vitro antibacterial effect test was conducted. Figure 1F shows that the inoculation area did not exhibit significant antibacterial zones. *C. gloeosporioides* continued to grow in this region, indicating that *B. amyloliquefaciens* GSBa-1 did not inhibit the growth and reproduction of *C. gloeosporioides* through spatial and nutritional competition. However, considering its effective inhibitory effect in vivo (Figure 1E), we hypothesized that this strain may induce effective resistance in mangoes, thereby suppressing the occurrence of anthracnose disease.

### 3.2. Impact of ABT on ROS Generation and Accumulation in Mangoes

Figure 2 illustrates the effects of ABT and on two ROS indicators in mangoes, including the H_2_O_2_ content and $O_{2}^{\cdot -}$ production rate. H_2_O_2_ and $O_{2}^{\cdot -}$ are critical intermediate products in ROS metabolism within plant cells. They can act as signaling molecules to enhance the activity of antioxidant pathways or potentially cause damage to plants [24]. As the storage period increased, the H_2_O_2_ content in the fruits also increased, with the H_2_O_2_ content being significantly higher in the peel than in the pulp (*p* < 0.05). In both the peel and pulp, the H_2_O_2_ content remained higher in the ABT group than that in the CKl group throughout the 12-day storage period. Both the peel and pulp exhibited significant peaks in $O_{2}^{\cdot -}$ production rates, with the pulp showing significantly higher levels than that in the peel. In the ABT group, the $O_{2}^{\cdot -}$ content in the peel reached a peak 9% higher than that in the CK group on the 4th day, while the $O_{2}^{\cdot -}$ content in the pulp peaked at 17% higher than that in the CK group on the 8th day. These findings indicate that ABT can induce substantial ROS production in mangoes during storage, potentially enhancing their resistance to external environmental stress.

### 3.3. Impact of ABT on ROS-Metabolizing Enzyme Activity in Mangoes

Figure 3 depicts the effects of ABT on the activity of ROS-metabolizing enzymes (SOD, CAT, APX, and POD) in the mango peel and pulp. Both the ABT and CK groups showed an increasing then decreasing trend in POD activity in the peel and pulp, which peaked on the 8th day. Additionally, ABT significantly increased POD activity in the mango peel and pulp. Similarly, ABT enhanced APX activity, with mango peel and pulp in the ABT group exhibiting significantly higher APX activity than those in the CK group throughout the 12-day storage period (*p* < 0.05). Furthermore, ABT had a more pronounced stimulatory effect on POD and APX activity in the mango pulp. On the 8th day, mango pulp from the ABT group exhibited a 17.4% and 37.9% increase in POD and APX activity, respectively, compared to the CK group. SOD and CAT activities showed significant peaks on the 8th day. Notably, before the 4th day, SOD activity was significantly higher in the CK group in both the mango peel and pulp, suggesting that ABT may initially inhibit mango SOD activity. However, on the 8th day, SOD activity in the peel and pulp of the ABT group was 62% and 24% higher, respectively, than those in the CK group. Unlike the other three enzymes, CAT activity in the mango peel was significantly higher after ABT compared to that in the pulp. These results indicate that ABT increased the activities of POD, APX, SOD, and CAT during the storage period. However, the stimulatory effects and timing of peak enzyme activities varied between the mango peel and pulp. This variation potentially promoted the continuous production and removal of ROS in mangoes over the 12-day storage period, effectively enhancing disease resistance by facilitating responses to external stimuli and reducing oxidative damage.

### 3.4. Impact of ABT on Enzyme Activities in the Phenylpropanoid Metabolic Pathway of Stored Mangoes

Figure 4 illustrates the effects of ABT on enzyme activities within the phenylpropanoid metabolic pathway in stored mangoes. PAL, C4H, 4CL, and C3H are key enzymes involved in this pathway. PAL and C4H activities in both the mango peel and pulp were significantly higher in the ABT group than in the CK group (*p* < 0.05). On the 8th day, the PAL and C4H activities in the mango peel and pulp of the ABT group were 84.5%, 56.1%, 49.6%, and 38.1% higher, respectively, than those in the CK group.

Throughout the storage period, the 4CL activity in the mango peel and pulp in the ABT group remained higher than that in the CK group. With extended storage times, the 4CL and C3H activities in the mango peel initially increased and then decreased. On the 8th day, these activities were 26.2% and 18.3% higher, respectively, in the ABT group than those in the CK group. Between the 8th and 12th days, the mango pulp in the ABT group exhibited significantly higher 4CL and C3H activities than that in the CK group (*p* < 0.05). On the 12th day, the activities of 4CL and C3H in the pulp in the ABT group were 49.1% and 7% higher, respectively, than those in the pulp in the CK group.

These changes in enzyme activities suggest that ABT stimulated the phenylpropanoid metabolic pathway more effectively in the mango peel. Thus, the phenylpropanoid pathway is likely more active in the peel, leading to the synthesis of phenolic compounds that enhance the defense capabilities of mangoes during storage.

### 3.5. Impact of ABT Treatment on the Accumulation of Phenolic Compounds in Stored Mangoes

Figure 5 depicts the effects of ABT on the content of major phenolic compounds in the mango peel and pulp during storage. Phenolic substances are crucial indicators of mango resistance and biological vitality. We identified the six predominant monomeric phenolic compounds in mangoes: gallic acid (Figure 5A), chlorogenic acid (Figure 5B), catechin (Figure 5C), ellagic acid (Figure 5D), ferulic acid (Figure 5E), and ursolic acid (Figure 5F). Among these, ellagic acid, gallic acid, and catechin were the primary phenolic compounds in the mango peel and pulp, with concentrations of approximately 1 mg/kg.

Throughout the storage period, gallic acid levels were significantly higher in the peel and pulp in the ABT group compared to those in the CK group (*p* < 0.05). On the 4th day, the gallic acid contents in the peel and pulp in the ABT group were 42.8% and 34.2% higher, respectively, than in those in the CK group. Moreover, throughout storage, the chlorogenic acid content in the mango peel in the ABT group remained consistently higher than that in the CK group (*p* < 0.05). On the 8th day, chlorogenic acid was undetectable in the pulp in the CK group, while that in the ABT group contained 0.0172 mg/kg chlorogenic acid. As the storage time increased, the catechin content in the mango peel decreased. The mango peel in the ABT group exhibited significantly higher catechin levels than that in the CK group (*p* < 0.05). There was no significant difference in pulp catechin content between the ABT and CK groups.

Between the 8th and 12th days, both the mango peel and pulp in the ABT group had significantly higher ellagic acid levels than those in the CK group (*p* < 0.05). On the 8th day, the ellagic acid content in the mango peel and pulp in the ABT group was 2.1 and 1.3 times higher, respectively, than that in the CK group. The ferulic acid content in the pulp in the ABT group was significantly higher than that in the CK group (*p* < 0.05), and it increased with storage time. On the 12th day, the pulp in the ABT group had 14.1% higher ferulic acid content than that in the CK group.

Between the 8th and 12th days, both the peel and pulp in the ABT group had significantly higher ursolic acid levels than those in the CK group (*p* < 0.05). On the 12th day, the ursolic acid contents in the mango peel and pulp after ABT treatment were 2.8 and 2.1 times higher, respectively, than those in the CK group.

These results mirror the enzymatic activities related to phenylpropanoid metabolism. ABT primarily stimulated the synthesis and accumulation of phenolic compounds, particularly gallic acid and ellagic acid, in the mango peel. This enhancement likely improves the defense capabilities of mangoes against adverse environmental stressors and disease resistance.

### 3.6. Correlation Analysis of Various Storage Quality Parameters in Mangoes during Storage

The experiments conducted in this study demonstrated that ABT treatment positively affected various storage quality parameters of mangoes during the post-harvest period, including disease incidence, disease severity index, and the activities of enzymes related to ROS metabolism. ABT enhanced mango disease resistance and reduced the incidence of anthracnose. However, the interrelationships among these parameters remain unclear. Therefore, we performed a correlation analysis on 18 datasets related to various changes during mango storage after ABT.

As shown in Figure 6, the correlation between disease incidence, disease severity index, ROS metabolism, and phenylpropane metabolism was stronger in the mango peel than in the fruit pulp. Disease incidence was strongly positively correlated with the peel H_2_O_2_ and $O_{2}^{\cdot -}$ contents, PAL activity, and 4CL activity. The disease severity index showed a strong positive correlation with the peel H_2_O_2_ content and 4CL enzyme activity. The $O_{2}^{\cdot -}$ content in the mango peel was strongly positively correlated with 4CL and C3H activity in the peel and PAL activity in the pulp. The H_2_O_2_ content in the peel was strongly positively correlated with 4CL activity in the peel and PAL and 4CL activity in the pulp.

The activities of enzymes related to ROS metabolism were strongly positively correlated with the peel PAL and C4H activities. C3H activity was also strongly positively correlated with the ROS-metabolizing enzymes, except for APX. Gallic and chlorogenic acid contents in both the peel and pulp were positively correlated with 4CL and C3H activities in the peel. Catechin content in both the peel and pulp were positively correlated with peel PAL and pulp C3H activities. Peel ferulic acid content was strongly correlated with PAL, C4H, and C3H activities. H_2_O_2_ and $O_{2}^{\cdot -}$ contents in the peel were strongly positively correlated with gallic acid, chlorogenic acid, ellagic acid, and ferulic acid levels in both the peel and pulp. Additionally, there was a strong positive correlation between catechin and ursolic acid levels in the peel. These findings reveal complex interactions between various quality parameters and indicate the multifaceted effects of ABT on mango storage quality during the post-harvest period.

## 4. Discussion

Anthracnose is a significant fungal disease caused by *Colletotrichum* spp. such as *C. gloeosporioides*. It leads to fruit rot in tropical fruits such as mangoes during post-harvest storage and transportation. *C. gloeosporioides* remains dormant until the mangoes ripen, and infection primarily occurs during the flowering period. Since anthracnose is a latent disease, many broad-spectrum antibacterial chemical preservatives have shown less effectiveness in controlling anthracnose in tropical fruits than expected [25,26]. In this study, ABT was effective in reducing disease incidence, proving its role in delaying maturity and maintaining mango quality. Although *B. amyloliquefaciens* GSBa-1 effectively inhibited anthracnose in mangoes, it did not significantly inhibit *C. gloeosporioides* in the in vitro antibacterial test. Therefore, it is likely that *B. amyloliquefaciens* GSBa-1 inhibits the incidence of anthracnose by enhancing the resistance of mangoes.

After ABT treatment, we observed a rapid increase in POD, APX, and CAT activities in mango peels, along with a significant increase in H_2_O_2_ levels. H_2_O_2_ accumulation in the peel was observed after storage for 4 days, and the superoxide anion generation in the pulp was significantly higher than that in the peel tissue after 4 days of storage. This was attributed to the non-biological stress response induced by the ABT on the mango peel, which facilitated its rapid response to external stimuli and accumulation of large amounts of ROS, primarily H_2_O_2_. H_2_O_2_ is an essential substance in the ROS defense mechanism of plants and acts as a signal molecule to activate active oxygen and other metabolic pathways [27]. However, excessive ROS accumulation can damage cell membranes. To counteract this, plants activate enzymatic and non-enzymatic ROS scavenging systems [28]. ABT significantly increased the activity of ROS-scavenging enzymes such as APX, SOD, and POD in mango pulp. Non-enzymatic antioxidant substances also play a role in removing main excess ROS. Zhang et al. [29] found that 1-methylcyclopropene treatment increased the hydrogen peroxide content in jujube fruits. Additionally, it significantly increased the activity of enzymes that remove ROS (APX and GSH) and enzymes involved in phenylpropanoid metabolism (PAL and 4CL).

The primary bioactive compounds found in fruits include phenolic acids, flavonoids, and anthocyanins. These polyphenolic compounds influence the storability and disease resistance of fruits through synthesis, accumulation, and degradation [30]. Research by Meng et al. [31] investigated the effects of cotreatment of 1-methylcyclopropylene and salicylic acid (SA) on mangoes and found that the primary mechanism of action involved stimulating disease resistance-related enzymes (phenylalanine ammonia-lyase, cinnamyl alcohol dehydrogenase, cinnamic acid-4-hydroxylase, p-coumaroyl-CoA ligase, peroxidase, polyphenol oxidase, chitinase, β-1,3-glucanase). This led to the accumulation of lignin, total phenols, and flavonoids, which effectively enhanced anthracnose resistance. Similarly, ABT in this study significantly enhanced active oxygen production and elimination, boosted the activity of enzymes associated with benzenepropanoid metabolism, and increased phenolic compound accumulation. The highest concentrations of phenolic acids in *Mangifera indica* Tainong No. l mangoes were ellagic acid, gallic acid, and catechins. These three primary phenolic acids exhibit distinct synthesis and degradation patterns in mango peel. Ellagic acid content in the peel peaked on the 8th day post-treatment, while gallic acid and catechins accumulated 4 days prior to storage. The synthesis and accumulation of these phenolic compounds may have been stimulated by H_2_O_2_. Furthermore, during the 8–12-day post-storage period, there was a notable disparity in the synthesis and accumulation of ursolic acid in the mango flesh in the ABT group, which was directly associated with the induction of superoxide anion in the fruit pulp. Similar activation of ROS metabolism and subsequent resistance induction was observed in a banana disease study, where ROS metabolism stimulated salicylic acid metabolism, inducing resistance to Fusarium wilt [32].

Figure 7 illustrates the integration of the pathways of active oxygen and synthetic phenolic compounds in benzenepropanoid metabolism. Based on the findings from this study, we inferred that *B. amyloliquefaciens* GSBa-1 enhances resistance to *C. gloeosporioides* by inducing active oxygen and benzenepropanoid metabolism in mangoes. When mango peels were exposed to *B. amyloliquefaciens* GSBa-1, a substantial amount of H_2_O_2_ was generated, primarily triggering the production of superoxide anions in the flesh. This process also directly stimulated C4H activity in the phenylpropanoid metabolic pathway. The superoxide anions in the flesh continued to activate upstream PAL and downstream 4CL and C3H activities, thereby inducing the synthesis and accumulation of gallic acid and ellagic acid in the peel and theaflavin in the flesh, consequently enhancing resistance against *C. gloeosporioides*. Similarly, Elsherbiny et al. [33] demonstrated that exogenous β-aminobutyric acid application enhanced resistance in citrus against *Penicillium digitatum* by enhancing the expression of disease-resistance-related proteomes and promoting phenylpropanoid metabolism and active oxygen metabolism. The post-harvest antibacterial mechanisms of antagonistic bacteria primarily include the secretion of antibacterial metabolites, spatial nutrient competition, hyperparasitism, and induction of host resistance [34]. In this study, *B. amyloliquefaciens* GSBa-1 effectively inhibited the growth of *C. gloeosporioides* in mangoes mainly by inducing resistance.

## 5. Conclusions

In this study, in vitro antimicrobial analysis revealed that *B. amyloliquefaciens* GSBa-1 did significantly inhibit the growth and reproduction of *C. gloeosporioides*. However, when applied as a surface coating on mangoes, *B. amyloliquefaciens* GSBa-1 demonstrated effective biocontrol activity by significantly suppressing the growth of *C. gloeosporioides* during mango storage, resulting in a 20% reduction in mango anthracnose incidence after 12 days of ambient temperature storage. This antagonistic bacterium enhances mango resistance by activating active oxygen and phenylpropanoid metabolism pathways. Specifically, *B. amyloliquefaciens* GSBa-1 induces the generation of ROS, such as hydrogen peroxide and superoxide anions, in the mango peel. This stimulation increases the activity of ROS-scavenging enzymes, including POD, SOD, and CAT. Additionally, *B. amyloliquefaciens* GSBa-1 boosts the activity of phenylpropanoid metabolism-related enzymes—PAL, C4H, 4CL, and C3H—in mango fruit flesh, leading to higher synthesis and accumulation of ellagic acid, gallic acid, and catechins. This accumulation of phenolic compounds aids in ROS removal and enhances resistance to anthracnose. Overall, the study elucidates the mechanism through which antagonistic bacteria inhibit *C. gloeosporioides* on mangoes.

## Figures and Tables

**Figure 1 metabolites-14-00417-f001:**
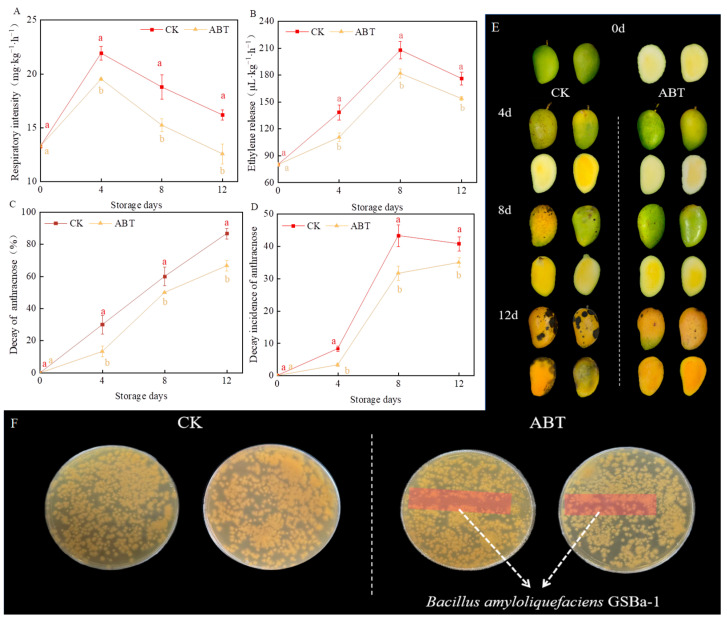
Effects of *Bacillus amyloliquefaciens* GSBa-1 treatment on mango respiratory intensity (**A**), ethylene release (**B**), decay rate of anthracnose (**C**), decay incidence of anthracnose (**D**), photos of the changes in the peel and pulp of mangoes during storage (**E**), and *Colletotrichum gloeosporioides* inhibition tests in vitro (**F**). (Note: In the figure, different lower case letters indicate significant differences in the same storage time (*p* < 0.05), the same as in the following figure).

**Figure 2 metabolites-14-00417-f002:**
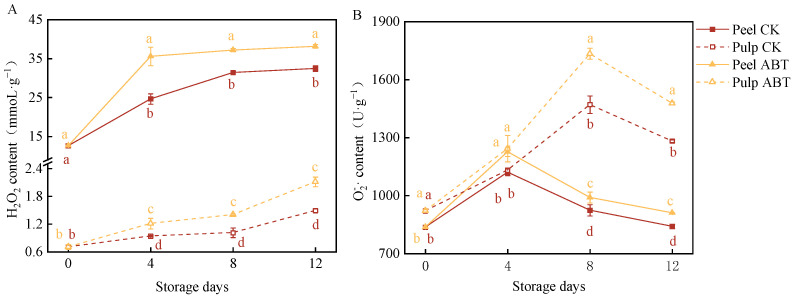
Effect of *Bacillus amyloliquefaciens* GSBa-1 treatment on the accumulation of mango H_2_O_2_ (**A**) and $O_{2}^{-}\cdot$(**B**).

**Figure 3 metabolites-14-00417-f003:**
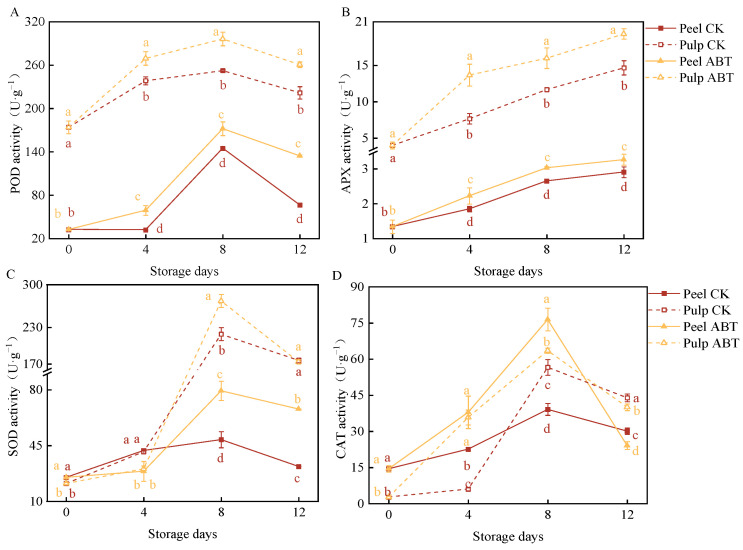
Effect of *Bacillus amyloliquefaciens* GSBa-1 treatment on activity of ROS metabolic enzymes POD (**A**), APX (**B**), SOD (**C**), and CAT (**D**) in mango. (Note: In the figure, different lower case letters indicate significant differences in the same storage time (*p* < 0.05)).

**Figure 4 metabolites-14-00417-f004:**
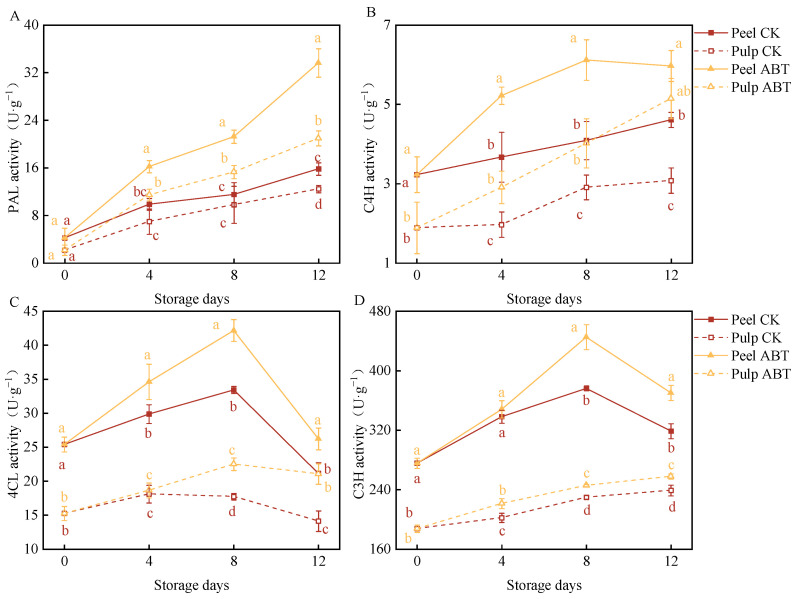
Effect of *Bacillus amyloliquefaciens* GSBa-1 treatment on the activity of the related enzymes PAL (**A**), C4H (**B**), 4CL (**C**), and C3H (**D**) in the mango phenylpropane metabolic pathway during the storage period. (Note: In the figure, different lower case letters indicate significant differences in the same storage time (*p* < 0.05)).

**Figure 5 metabolites-14-00417-f005:**
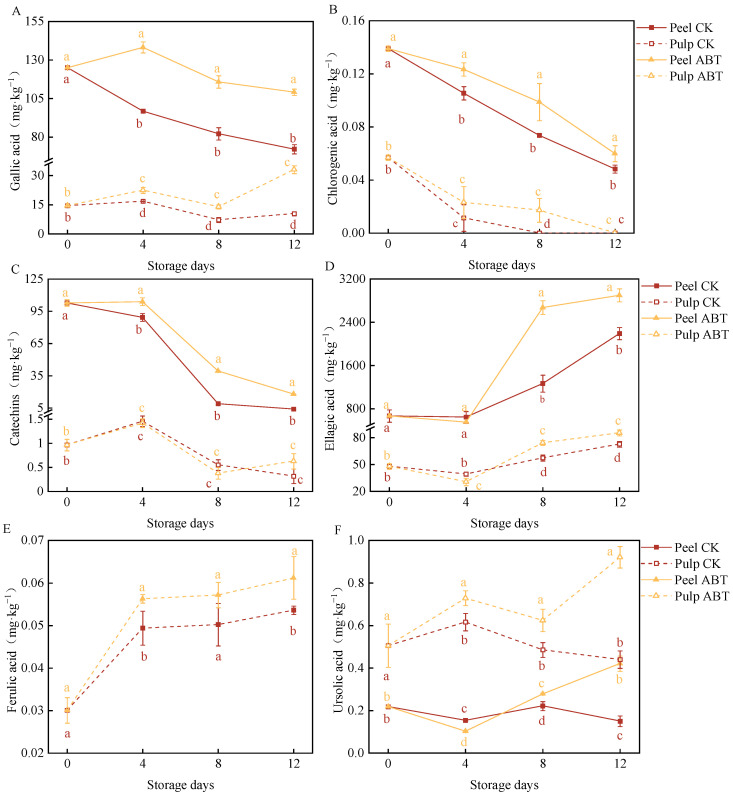
Effect of *Bacillus amyloliquefaciens* GSBa-1 treatment on the content of phenolic compounds gallic acid (**A**), chlorogenic acid (**B**), catechins (**C**), ellagic acid (**D**), ferulic acid (**E**), and ursolic acid (**F**) in mango. (Note: In the figure, different lower case letters indicate significant differences in the same storage time (*p* < 0.05)).

**Figure 6 metabolites-14-00417-f006:**
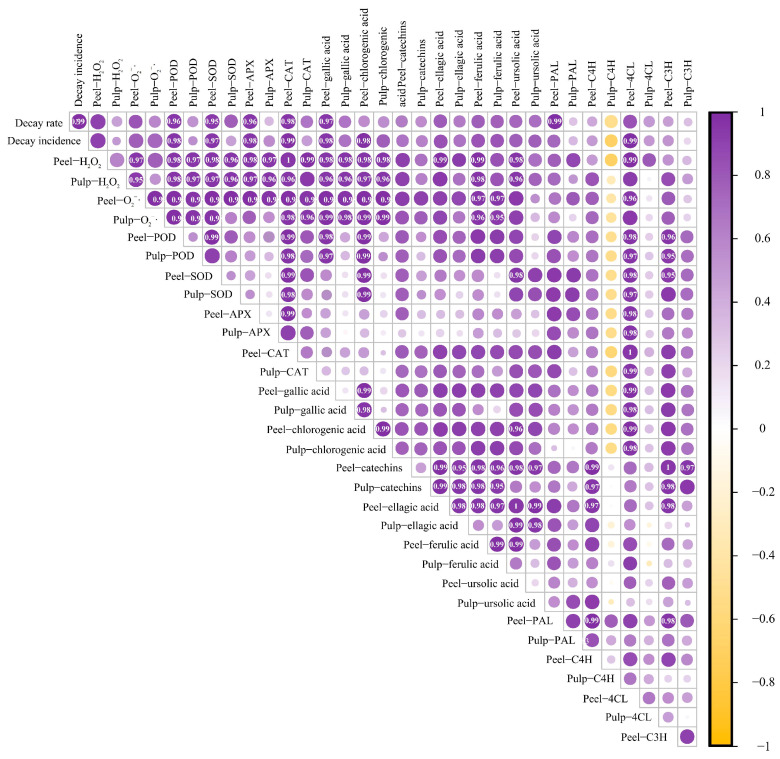
Correlation analysis of various storage quality indicators during the mango storage process. (Note: Marked values indicate the correlation coefficient, the stronger the correlation, greater than 0.8 is the strong correlation).

**Figure 7 metabolites-14-00417-f007:**
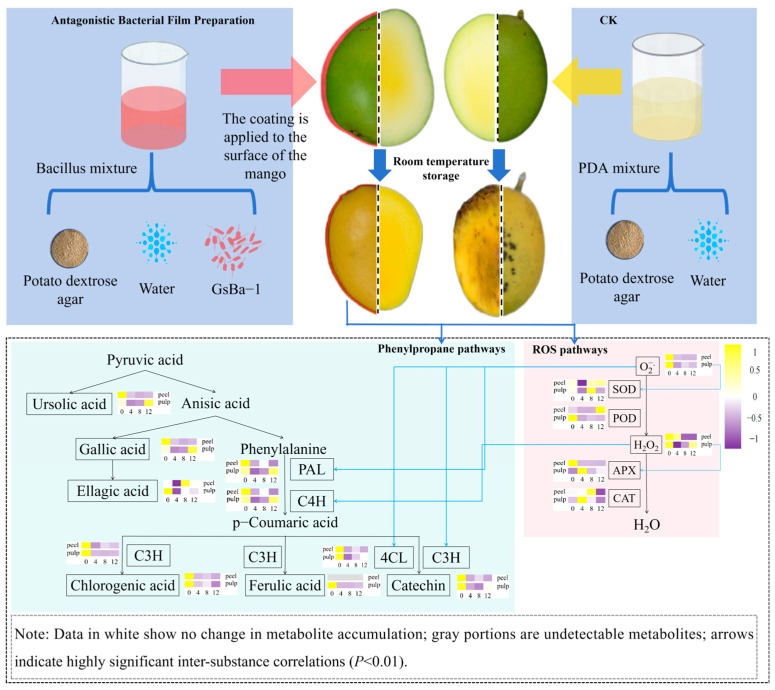
Mechanism hypothesis of ROS and phenylpropane metabolism induced by *Bacillus amyloliquefaciens* GSBa-1 antagonistic membrane treatment in mango.

## Data Availability

The data used to support the findings of this study can be made available by the corresponding author upon request.

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
