# Peer review of "Improving Resistance of Mango to Colletotrichum gloeosporioides by Activating Reactive Oxygen Species and Phenylpropane Metabolism of Bacillus amyloliquefaciens GSBa-1"

_metabolites, 2024, doi:10.3390/metabo14080417_

Round 1
Reviewer 1 Report
Comments and Suggestions for Authors
1- objective of the study are not clear rewrite again.
2- check the suitable statistical design and check the significancy
3- disscusion did not support the result
4. Rewrite the conclusion
Comments on the Quality of English LanguageExtensive editing of English language required
Reviewer 2 Report
Comments and Suggestions for Authors
Manuscript review of metabolites-3085963
Study on improving resistance of mango to anthracnose by activating active oxygen species and phenylpropane metabolism of Bacillus amyloliquefaciens GSBa-1
Wenya Li, Hua Chen, Jianhu Cheng, Min Zhang , Yan Xu, Lihua Wang, Xueqiao Zhao, JinYao Zhang, Bangdi Liu , and Jing Sun
Currently, the mechanisms and treatments that increase plant resistance to abiotic environmental factors are fairly well presented in the literature. However, plant defense against pathogens is poorly studied. The study of preservation of food products (mangoes) during storage is particularly needed.
The manuscript is timely because the results of the study allow us to expand the ideas about the mechanisms of resistance to pathogens and to propose an environmentally friendly way to protect Mangifera indica L. fruits from pathogens, namely using in this case exogenous treatment with Bacillus amyloliquefaciens GSBa-1.
The authors investigated the mechanism of action of Bacillus amyloliquefaciens GSBa-1 on the formation of anthracnose resistance of mango fruits. They found a direct dependence of mango resistance to anthracnose on the activation of ROS, antioxidant enzymes and enzymes of phenolic metabolism. They showed an advantage in the formation of anthracnose resistance in the peel of mango fruit compared to its pulp.
Remarks.
1. Please remove from the title "Study..."
2. Please check the correct use of the word "anthrax " on pages 1 (l 28), 4 (l 155), 6 (l 146) and 7 (l 255).
3. Please change the keywords. They should specify the problem being studied.
The Name of the bacteria and the plant under study should be entered in Latin.
Bacillus amyloliquefaciens; Mangifera indica; ROS; antioxidant enzymes; anthracnose resistance; phenylpropanoid metabolic pathway.
4. Page 7. Figure 1: No space between word and parenthesis (Ð .7., l 255 and Ð .7., l 256)
5. Please replace the word "viability" with the word "activity" in the caption to Figures 3 and 4 (Page 9, l. 305 and Page 10, l. 330).

Author Response
Comment 1:Please remove from the title "Study...”
Response: We appreciate reviewer’s suggestion. We have removed the "Study on" from the original title. The new title is now "Improving resistance of mango to anthracnose by activating reactive oxygen species and phenylpropane metabolism of Bacillus amyloliquefaciens GSBa-1".
Comment 2: Please check the correct use of the word "anthrax"on pages 1 (L 28),4(L 155),6(L 146) and7(L 255).
Response: We appreciate reviewer’s suggestion. We carefully examined the use of the word "anthrax" on page 1 (L 28), page 4 (L 155), page 6 (L 146), and page 7 (L 255) and throughout other parts of the manuscript. The incorrect word "anthrax" has been replaced with the "anthracnose" or the specific Latin name of the fungal strain (" Colletotrichum gloeosporioides").
Comment 3: Please change the keywords. They should specify the problem being studied. Bacilus amyloliquefaciens, Mangifera indica, ROS, antioxidant enzymes, anthracnoseresistance, phenylpropanoid metabolic pathway.
Response: We appreciate reviewer’s suggestion. We have revised five key words to ensure that they are more closely aligned with the main research topic of this paper. The revised key words are "Mangifera indica", "antagonistic bacteria preservation", "resistance to anthracnose", "reactive oxygen species" and "phenylpropanoid metabolic pathway".
Comment 4: The Name of the bacteria and the plant under study should be entered in Latin
Response: We appreciate reviewer’s suggestion. We have revised the names of the antagonistic bacteria, anthracnose, and mango used in the article to their Latin names, which is Mangifera indica Tainong No.l mangoes, Bacillus amyloliquefaciens GSBa-1 and Colletotrichum gloeosporioides.
Comment 5: Page 7. Figure 1: No space between word and parenthesis (P.7., I 255 and P.7., I 256)
Response: We appreciate reviewer’s suggestion. We checked and revised the space between words and parentheses in Figure 1 on page 7.
Comment 6: Please replace the word "viability" with the word "activity" in the caption to Figures 3 and 4(Page 9,I. 305 and Page 10.I. 330).
Response: We appreciate reviewer’s suggestion. We have revised the word "viability" in the title of Figures 3 and 4 into "activity" (page 9 L 305 and page 10 L 330).
Reviewer 3 Report
Comments and Suggestions for Authors
1. The manuscript has mentioned reactive oxygen instead of active oxygen species. Correct accordingly
2. The value of CFU is not mentioned properly
3. Abstract has mentioned “anthrax” and also in other places
4. Line 46: Production quantity of 2022 can not be expected production. Please recheck and revise
5. Scientific names should be given in italics
6. Instead of water, the medium used in culturing bacteria needed to be used.
7. Methodology of the treatment of fruits and making the culture are not properly mentioned in the paper
8. Other portions of the methodology need a revision for grammatical mistakes.
9. Line 166: “suspension of conidio spores of Bacillus anthracis onto the plates” Why this bacteria was chosen for invitro study?
10. Fig 1F : doesn’t convey the meaningful results
11. Reference 10 has discussed about only Buddha's hand fruit, but line 74 mentioned about mangoes too. The present study is just change in fruit crop, and adding the fungal disease aspect. The mechanism of delayed ripening is already published in reference 10. In the present study, the effect of ABT on ripening is not discussed. Without fungal pathogen, the effect of ABT on mango is not studied
Comments on the Quality of English Languagena
Author Response
Comment 1: The manuscript has mentioned reactive oxygen instead of active oxygen species. Correct accordingly
Response: We appreciate reviewer’s suggestion. We have reviewed the entire manuscript and replaced "active oxygen species" with "reactive oxygen".
Comment 2: The value of CFU is not mentioned properly
Response: We appreciate reviewer’s suggestion. In previous studies, only the concentration of the bacterial suspension used for dipping and coating mangoes was mentioned at 1*108 CFU/mL, but no other conditions of dipping and coating were mentioned. We have added the amount of bacterial suspension solution used for dipping and the ratio of liquid to solid ingredients to the method section.
Comment 3: Abstract has mentioned "anthrax" and also in other places
Response: We appreciate reviewer’s suggestion. We have reviewed the entire manuscript and have revised "anthrax" to the correct term "anthracnose" or the Latin name "Colletotrichum gloeosporioides".
Comment 4: Line 46: Production quantity of 2022 can not be expected production. Please recheck andrevise
Response: We appreciate reviewer’s suggestion. This is a spelling error that has been corrected. We have revised the year 2022 to 2020. These data were obtained from Reference 1 and were taken from the mango production data in FAO.
Comment 5: Scientific names should be given in italics
Response: We appreciate reviewer’s suggestion. We have revised the names of the antagonistic bacteria, anthracnose, and mango used in the article to their Latin names, which is Mangifera indica Tainong No.l mangoes, Bacillus amyloliquefaciens GSBa-1 and Colletotrichum gloeosporioides.
Comment 6: Instead of water, the medium used in culturing bacteria needed to be used.
Response: Thank you for your detailed comments. It is true that the solution used for the coating in the study was not water, but PDA medium within Bacillus amyloliquefaciens GSBa-1. We have revised in the Methods section and Figure 7 to avoid misleading readers about the treatment method of our article.
Comment 7: Methodology of the treatment of fruits and making the culture are not properly mentioned in the paper
Response: We appreciate reviewer’s suggestion. We have added a description of the coating treatment and storage conditions for mangoes in the Methods section.
Comment 8: Other portions of the methodology need a revision for grammatical mistakes
Response: We appreciate reviewer’s suggestion. We have carefully read and revised the method section of our paper, and we have also had a native English speaker proofread the entire paper for grammar to ensure that the intended meaning is accurately conveyed.
Comment 9: Line 166: “suspension of conidio spores of Bacillus anthracis onto the plates" Why this bacteria was chosen for invitro study?
Response: Thank you for the reviewer's questions. Here we chose to apply anthracnose (Colletotrichum gloeosporioides) on PDA medium for in vitro antibacterial test to verify whether Bacillus amyloliquefaciens GSBa-1 as a biocontrol agent can inhibit Colletotrichum gloeosporioides through spatial and nutritional competition. We have added the purpose of this experiment to Method 2.3.2.
Comment 10: Fig 1F:doesn't convey the meaningful results
Response: Thank you for reviewers' comments. We have revised the description of the results in Result 3.1 for Figure 1 to ensure a more accurate representation of the experimental content.
Comment 11: Reference 10 has discussed about only Buddha's hand fruit, but line 74 mentioned about mangoes too. The present study is just change in fruit crop, and adding the fungal disease aspect. The mechanism of delayed ripening is already published in reference 10. In the present study, the effect of ABT on ripening is not discussed. Without fungal pathogen, the effect of ABT on mango is not studied
Response: Thank you for the reviewer's questions. We have added reference 11 after reference 10. In reference 11, we report on our previous studies that show Bacillus amyloliquefaciens GSBa-1 can inhibit the internal infection diseases of mango and winter melon.
Round 2
Reviewer 1 Report
Comments and Suggestions for Authors
The Manuscript was revised dramatically. Now accept the manuscript in a present form.
Comments on the Quality of English LanguageMinor editing of English language required carefully check the typos in the article.
Reviewer 3 Report
Comments and Suggestions for Authors
Authors have addressed all the points raised , and can be accepted